# A Preliminary Study on the Meaning of Inflammatory Indexes in MS: A Neda-Based Approach

**DOI:** 10.3390/jpm13111537

**Published:** 2023-10-26

**Authors:** Sena Destan Bunul, Aybala Neslihan Alagoz, Bilge Piri Cinar, Fatih Bunul, Seyma Erdogan, Husnu Efendi

**Affiliations:** 1Department of Neurology, Faculty of Medicine, Kocaeli University, Kocaeli 4100, Turkey; aybala.alagoz@kocaeli.edu.tr (A.N.A.); seyma.yilmazguc@kocaeli.edu.tr (S.E.); husnu.efendi@kocaeli.edu.tr (H.E.); 2Department of Neurology, Faculty of Medicine, Samsun University, Samsun 5500, Turkey; bilge.cinar@samsun.edu.tr; 3Internal Medicine, Anadolu Medical Center, Kocaeli 4100, Turkey; fatih.bunul@anadolusaglik.org

**Keywords:** multiple sclerosis, inflammation, biomarkers

## Abstract

Background: Multiple sclerosis (MS) is a disease of the central nervous system characterized by inflammation, demyelination, and axonal degeneration. This study aimed to investigate the relationship between inflammatory indexes and MS disease activity and progression. Methods: A prospective cohort study was conducted at the Kocaeli University Neurology Clinic, involving 108 patients diagnosed with MS. Data related to patient demographics, clinical presentations, radiological findings, and laboratory results were recorded. Inflammatory markers such as NLR (neutrophil-to-lymphocyte ratio), PLR (platelet-to-lymphocyte ratio), MLR (monocyte-to-lymphocyte ratio), and indexes such as SII (systemic immune inflammation index), SIRI (systemic immune response index), and AISI (systemic total aggregation index) were examined to determine their correlation with MS disease activity and disability. When assessing the influence of SII, AISI, and SIRI in predicting NEDA, it was found that all three indexes significantly predict NEDA. All indexes demonstrated a significant relationship with the EDSS score. Notably, SII, SIRI, and AISI were significant predictors of NEDA, and all inflammatory indexes showed a strong intercorrelation. This study investigates the role of inflammation markers in MS patients. It suggests that one or more of these non-invasive, straightforward, and practical markers could complement clinical and radiological parameters in monitoring MS.

## 1. Introduction

Multiple sclerosis (MS) is a complex neurological disease that affects an estimated 2.5 million people worldwide and has significant socio-economic consequences. Inflammation plays an important role in disease progression and symptom onset in MS, a central nervous system (CNS) disease characterized by demyelination and axonal disruption. The origins of the disease are multifaceted, and its pathogenesis relies on a combination of genetic, environmental, and immunologic factors. The complex nature of its causes makes MS a challenging condition to understand and manage, emphasizing the need for extensive research and targeted therapeutic interventions [1]. 

This complexity not only complicates our full understanding of the disease’s pathogenesis but also poses one of the greatest challenges to achieving successful treatment outcomes. The primary cause of the inflammatory processes observed in multiple sclerosis is believed to be the peripheral activation of Th1 cells that interact with myelin autoantigens and then migrate to the central nervous system. Upon activation, these T-cells attract various myeloid cells, including monocytes and macrophages, triggering an inflammatory response that often results in axonal transection and irreversible localized CNS damage [2].

Therapeutic targets in the MS field are continuously evolving due to recent advances in disease-modifying treatments (DMTs) [3]. It is widely accepted that the primary treatment goal in all stages of MS should be to reduce relapses, lesions, and brain atrophy. This perspective is particularly important in the early stages of MS, when it is possible to minimize the number of new lesions and the extent of brain inflammation, both of which lead to atrophy. In light of current knowledge, the current goal is to achieve complete clinical and radiologic remission of the disease rather than simply aiming to reduce relapse rates and disability progression [4]. Over the last decade, evidence has pointed to a window of opportunity for successful treatment of MS patients that spans the period of peak CNS inflammation. Since MS treatments are particularly effective against inflammatory processes, it is believed that timely and effective therapy could delay or reduce the neurodegenerative process [5]. The absence of relapses, disability progression, or magnetic resonance imaging (MRI) activity characterizes “no evidence of disease activity” (NEDA-3). NEDA-3 composite measurement has emerged as a new therapeutic target for MS patients thanks to the easy availability of novel, highly effective immunotherapies [6].

A complete blood count (CBC) test measures the number and morphology of various cell types in the blood. White blood cells, neutrophils, lymphocytes, erythrocyte sedimentation rate, and C-reactive protein (CRP) are commonly used markers for evaluating inflammation. While these characteristics can indicate the presence and degree of inflammation, they alone are not sufficient for a definitive diagnosis. Ratios derived from hemogram parameters, such as neutrophil-lymphocyte ratio (NLR), platelet-to-lymphocyte ratio (PLR), and monocyte-to-lymphocyte ratio (MLR), have been used as inflammation indicators in numerous diseases [7,8,9,10]. 

The systemic immune-inflammatory index (SII), which comprises neutrophil, lymphocyte, and platelet counts, has recently been recognized as a new measure of inflammation. SII can serve as a biomarker in various conditions and may indicate systemic inflammation. According to research, SII has been associated with MS disease activation, MRI activation, and disability [11,12]. Other metrics, such as the systemic immune response index (SIRI) and the systemic total inflammation index (AISI), can also serve as inflammatory biomarkers [13,14]. However, no published studies have used SIRI or AISI in the context of MS. 

The idea of identifying easy and practical biomarkers and adding other markers to patient follow-up and monitoring has attracted much attention in the last decade. However, biomarkers that are potentially more noninvasive for patients and that can be obtained from routine examinations may be more promising than expected. 

The aim of this study was to explore the relationship between easily accessible, cost-effective, and non-invasive serum inflammatory biomarkers and MS disease activity and disability. Additionally, the predictive value of the inflammatory parameters (SII, SIRI, and AISI) for NEDA was examined.

## 2. Materials and Methods

This prospective cohort study was conducted from April 2021 to May 2023. Patients who achieved NEDA-3 by the end of the second year following the initial evaluation were included in the study. Over the course of two years, patients underwent examinations every six months by one blinded neurologist MS specialist. An MRI was performed annually, ensuring that each patient had a minimum of three MRI evaluations at baseline, after the first year, and after the second year. MRI findings were assessed by two blinded neurologists. The ethics committee of the Kocaeli University Faculty of Medicine approved the study (GOKAEK-22/06.11). Informed consent was obtained from each patient for the study.

### 2.1. Data Collection

#### 2.1.1. Participitians

A total of 108 MS patients followed in the Kocaeli University Neurology Clinic were included in the study. Inclusion criteria were that patients must have been diagnosed with relapsing-remitting MS, have been on the same medication for at least a year, be 18 years or older, and meet the 2017 revised McDonald criteria. Patients were excluded for neurological or psychiatric illnesses, chronic conditions such as thyroid disorders, coronary artery disease, diabetes mellitus, malignancy, a recent history of infection, or if they had an attack or received corticosteroid treatment in the last three months. All participants included in the study were selected from patients who had used a DMT (a group whose treatment had not changed in the last year) for at least one year. The patients’ demographic data, such as age, gender, and body mass index (BMI), and their clinical, radiological, and laboratory evaluations were recorded in a case report form. EDSS scores were calculated following their neurological examinations. All participants underwent the timed 25-foot walk test (T25FWT), the 9-hole peg test (NHPT), and the symbol digit modalities test (SDMT). The T25FW test was used to assess lower extremity function. In this test, the time taken by the patient to cover a distance of 25 feet (approximately 8 m) is measured. The test is repeated twice and averaged. Upper extremity function was assessed using the NHPT. This test involves inserting pegs into nine holes as quickly as possible. The test is repeated four times, twice for each hand, and an average result is obtained. SDMT is a short, easy to administer, and understandable test used to assess cognitive status and especially information processing capacity in neurological diseases. In the SDMT, the patient is asked to write the digit specified in the proposition under each figure as fast as possible in 90 s by looking at the proposition in which certain geometric shapes are paired with certain numbers. Responses can be written or verbal. At the end of 90 s, the number of correct and incorrect answers is recorded. Additionally, the DMTs taken by the patients were recorded. For disease activity assessment, the study group was categorized according to NEDA-3 status at the end of the second year. NEDA’s sub-parameters were defined as relapse, MRI activity (the presence of a new T2 lesion or of a gadolinium-enhancing lesion, evaluated separately), and EDSS progression. At the end of the second year, blood samples were collected within two weeks from patients who achieved NEDA-3. Patients who had a history of infection in the last two weeks, those who had used anti-inflammatory drugs, antibiotics, or antiviral medications in the previous two weeks, patients currently menstruating, and patients who had taken corticosteroids in the last three months were excluded.

#### 2.1.2. MRI Acquisition

All MRI examinations were performed using the same MRI device (Philips Ingenia, 3Tesla). In the MRI examination, T1-weighted sequences, T1-weighted sequences with gadolinium, T2-weighted sequences, and fluid-attenuated inversion recovery (FLAIR) sequences were utilized. 

#### 2.1.3. Serum Samples

All venous blood samples were collected using potassium ethylenediaminetetraacetic acid (EDTA) tubes (Beckman Coulter UniCel D × H 800). The CBC was assessed within one hour of collection using a blood cell analyzer. The Sysmex XN-1000 series blood cell analyzer has adopted three key technologies for blood cell analysis: semiconductor laser flow cytometry, which detects structural changes in cells; direct current sheath flow measurement method used to count red blood cells and platelets; sodium lauryl sulfate (SLS)-hemoglobin method, which measures hemoglobin concentration.

In addition, surfactants and markers were used to differentiate cells based on their marking intensity and membrane resistance. The reagents required for these processes were provided by companies such as Becton Dickinson and Company of Japan, Stemcell Technologies, and Hitachi High-Technologies. The samples were tested for lymphocyte, monocyte, platelet, and neutrophil levels. All serum samples were collected between 08:00 and 09:00. The calculation formulas for all inflammation indexes included in the study are provided in Table 1.

### 2.2. Statistical Methods

The Windows Statistical Package for Social Sciences (SPSS) version 22.0 was used for all statistical analyses to evaluate the study data. A *p*-value of less than 0.05 was considered statistically significant. The normality of the data distribution was assessed using the Shapiro–Wilk test, and by examining histograms and plots. For data with a normal distribution, the mean and standard deviation values are provided. For data without a normal distribution, the median value (min–max) is provided. Frequencies in categorical data are presented as percentages. The nonparametric Mann–Whitney U test was used to compare the means of two independent groups with non-normally distributed data, while the student *t*-test was used for normally distributed data. To compare the means between more than two groups, the nonparametric Kruskal–Wallis analysis of variance was used. The Spearman correlation test determined the direction and strength of the relationships between variables. The effects of SII, AISI, and SIRI in predicting NEDA in the study group were analyzed using binary logistic analysis.

## 3. Results

### 3.1. Demographic and Physical Features of the Study Group 

A total of 108 MS patients were included in the study, of which 71 (65.7%) were women. The average age was 37.45 ± 9.36 years. The median disease duration in the study group was 65 months (13–288); median EDSS score 1.5 (0.5–5); median BMI value 25 (15.70–33.20); mean SDMT score 34.80 ± 12.48; median NHPT (D) time 22.07 s (15.04–60.97); median NHPT (L) time 23.67 s (14.71–60.51); and median T25FWT time 11.79 s (2.50–35.00).

### 3.2. Hemogram Parameters and Inflammation Indices

Hemogram values, biochemical data, and the indexes derived from this data for the study group are presented in Table 2. 

### 3.3. NEDA-3 Status

When evaluating the study group based on NEDA status, 50% of the patients (54/108) achieved NEDA. The EDSS scores of the NEDA-achieving group were significantly lower than those who did not (1.04 ± 0.94 versus 2.6 ± 1.47; *p* < 0.001). There was no difference in terms of age and gender between the group that achieved NEDA-3 and the group that did not. Inflammation indexes were higher in the group that achieved neda-3 than in the group that did not achieve neda-3. Significant findings based on the presence of NEDA are provided in Table 3. 

When the group not achieving NEDA was evaluated in terms of NEDA subparameters, EDSS progression was observed in 48.1% (26/54) of the group. On the other hand, 10 patients (18.5%) had a new T2 lesion, and 9 (16.6%) had a Gd+ lesion. As a result, while MRI activation was observed in 19 patients (35.1%), relapse was observed in only 4 (7%) of the group that did not achieve NEDA (Figure 1).

### 3.4. Correlation Analyses 

Upon examining the relationship between the study group’s demographic characteristics and their physical and cognitive parameters, a mild correlation was found between age and both NHPT (D) and NHPT (L) times (rho = 0.219 and 0.255; *p* = 0.023 and *p* = 0.008, respectively). A slight negative correlation was also observed between age and SDMT (rho = −0.243; *p* = 0.011). Correlations between physical parameters and inflammation markers are presented in Figure 2 and Figure 3.

When examining the correlations among the inflammation indexes assessed in this study, all indexes were found to be strongly interrelated. Correlation values are provided in Table 4.

### 3.5. Assessment of NEDA Status and Inflammation Indexes

Differences in SII, SIRI, AISI, and lymphocyte values between treatment groups were evaluated using the Kruskal–Wallis test. A significant difference was observed only for lymphocyte counts, whereas no statistically significant differences were detected for SII, SIRI, and AISI (*p* = 0.022, *p* = 0.132, *p* = 0.394, and *p* = 0.522, respectively). In the study group, the effectiveness of SII, AISI, and SIRI in predicting NEDA was assessed with binary logistic analysis. All three indexes were found to significantly predict NEDA. Age, BMI, and gender showed no significant effect. An increase of one unit in AISI and SII values raised the likelihood of achieving NEDA by 1.001 times, while a one-unit increase in SIRI increased this likelihood by 1.51 times (*p* < 0.001, *p* = 0.001, and *p* < 0.001, respectively) (Table 5).

When lymphocyte values in the study group were examined according to all drugs, as expected, significantly lower lymphocyte values were detected in the fingolimod group compared to the other drug groups (*p* < 0.001). However, the DMD groups were compared using the Kruskal-Wallis test in terms of SII, SIRI, and AISI values; no significant relationship was detected between the three indexes and lymphocyte values (*p* > 0.05) (Table 6).

## 4. Discussion

There have been many studies on the potential of inflammatory biomarkers to predict disease activity or prognoses in the monitoring of MS. This study investigated the overall efficacy of inflammatory biomarkers for disease activity, physical disability, and cognitive impairment. In our research, the inflammatory indexes were found to be higher in the NEDA-3 group. While there was a significant relationship between EDSS, T25FWT, and inflammatory indexes, no association was observed between NHPT and SDMT. The correlations among the inflammatory indexes displayed very strong positive relationships with one another. Moreover, we assessed the predictive value of the inflammatory indexes for NEDA-3 status (for 2-year NEDA) and found that all three parameters (SII, SIRI, and AISI) were strong predictors of NEDA-3.

In general, the functions of the upper extremities and limbs can be influenced by age. Older MS patients may take slightly longer to complete NHPT and T25FWT compared to younger individuals [15,16]. Consistent with expectations, our study identified a positive correlation between NHPT, T25FWT, and age. SDMT is the most prevalent cognitive test for MS patients. Our study revealed a negative correlation between SDMT scores and age. This decline is linked to the deterioration of cognitive abilities due to aging in MS patients, compounded by the effects of neurodegeneration over time [17]. Furthermore, research indicates a relationship between physical and cognitive functions, with cognitive impairment being more prevalent in patients with high EDSS scores [18]. In line with prior studies, our research found a significant negative correlation between SDMT and EDSS.

In MS, the inflammation and neurodegenerative processes are complex. Recently, serological inflammatory indexes, combined with clinical, laboratory, and radiographic indicators, have been studied. Comprehensive investigations, including serum biomarkers, can offer insights into subclinical inflammation [19,20]. While the overall leukocyte count is a reliable indicator of inflammation, it does not consider the varied kinetics of different leukocyte subsets and can be affected by numerous factors. Integrative immune-inflammatory markers, such as NLR, PLR, SII, AISI, and SIRI, have been introduced in recent years to shed light on the connection between inflammation and autoimmune diseases [21,22,23].

The neutrophil-to-lymphocyte ratio (NLR) is a simple, rapid, nonspecific, and cost-effective method for identifying systemic inflammation. Compared to healthy controls, individuals with autoimmune diseases, such as Sjogren’s syndrome, sarcoidosis, ulcerative colitis, rheumatoid arthritis, and MS, have shown higher NLR values [7]. Numerous studies have compared NLR levels in MS patients and healthy controls, and the majority indicate that NLR levels are elevated in MS patients [8,24,25,26]. A meta-analysis of NLR as an inflammation marker in MS determined that NLR is higher in MS patients than in healthy controls, more elevated during relapse than in remission, and showed no difference in NLR among different clinical phenotypes of MS [27]. In our study, NLR values correlated with T25FWT and EDSS. Additionally, the NLR value was higher in the NEDA-3 group. Given this information, while NLR might serve as a valuable biomarker for MS, it remains unclear whether it is more indicative of the inflammatory or progressive stages of the disease.

PLR is another immunological inflammatory marker. Unlike NLR, PLR accounts not only for leukocyte subsets but also for platelet numbers [9]. One study found that an increase in PLR in MS patients might serve as an inflammatory marker. Moreover, a small yet statistically significant association was observed between increased PLR values and EDSS scores [11]. MLR is a novel biomarker for MS monitoring. There are few studies on MLR and MS. Hemond et al. found that a higher MLR significantly predicts MS progression, irrespective of demographic, clinical, or psychosocial factors. In this study, both PLR and MLR, similar to NLR, were found to be significantly higher in the NEDA-3 group.

While NLR and PLR are determined by the ratios of two distinct blood cell populations, the SII index combines all three populations by multiplying NLR with platelet counts [9]. One study showed that SII levels were higher in MS patients, both with and without radiologically active lesions, compared to healthy controls. Furthermore, SII levels were notably higher in patients with active lesions than in those without. Due to its integration of information from three blood parameters, SII offers superior sensitivity and specificity in MS patients [11]. In another study, SII displayed the strongest association with EDSS compared to other inflammatory indicators, suggesting its predictive value for disability [12]. No existing literature has explored the relationship between NEDA and SII, AISI, and SIRI. Furthermore, there has been no research on SIRI and AISI in MS patients. Our study is the first to examine SIRI and AISI in MS patients, finding that all three indexes are significant in predicting NEDA-3 status. We observed a notable association between SII, SIRI, AISI, and EDSS. This correlation underscores the internal consistency of the indexes and the potential of SII, SIRI, and AISI as emerging biomarkers in MS.

In recent years, biomarkers such as kappa immunoglobulin free light chain concentration, kappa cerebrospinal fluid (CSF) index proposed as an alternative to oligoclonal bands, neurofilament light chain, glial fibrillary acidic protein, and myelin basic protein have been extensively studied in CSF and serum to assess disease activation and progression in MS [28]. However, the limited reproducibility of data obtained from CSF, the high cost, technical difficulties, and standardization problems of new biomarkers in serum limit the widespread use of these markers. Our study focuses on inflammatory markers derived from serum, highlighting their advantages of easy accessibility, low cost, practicality, and especially reproducibility.

In recent years, there has been extensive discussion about the use of NEDA—the absence of evidence of disease activity—in follow-up as a measure to evaluate therapy response. In this study, when comparing groups that achieved NEDA-3 to those that did not, all inflammatory parameters were found to be significantly higher in the group that achieved NEDA-3 for two years. This raises questions about the true definition of NEDA. There may be several reasons why many neurologists are cautious about using NEDA in clinical practice. While NEDA is a comprehensive indicator, its sub-parameters, including clinical relapses, disability progression, and MRI activity, can be difficult to interpret reliably. There are concerns that reliance on NEDA alone may miss subtle but important changes in a patient’s health. Moreover, strict NEDA criteria may not capture the full spectrum of disease variations, potentially leaving gaps in treatment approaches [29,30,31,32]. While the NEDA group is, by definition, believed not to exhibit disease activity, our findings suggest that inflammatory processes may persist in these patients. The fact that the NEDA group was more inflammatory can be explained by the fact that almost half of the group that did not achieve NEDA in our study showed EDSS progression, as seen in Figure 1. Our study observed a decrease in inflammatory parameters during the progression of MS, paired with an increase in EDSS and the predominance of the neurodegenerative process, aligning with previous literature. This suggests that as MS progresses, the inflammatory phase may subside, giving way to a more neurodegenerative phase. The shift in disease dynamics underscores the importance of timely interventions and personalized therapeutic strategies. Furthermore, understanding these transitions could provide valuable insights into the mechanisms underlying MS and guide future research towards the development of more effective treatments [33,34,35,36]. 

In the current study, when the indexes and lymphocyte values were examined based on the DMTs used by the patients, no significant relationship was established between the indexes and lymphocyte counts. This may refute the view that drugs that cause lymphopenia can affect the inflammation index.

### Limitations

One limitation of this study is that although factors potentially affecting systemic inflammation within the study group were controlled, it must be acknowledged that these indexes could be influenced by factors such as disease-modifying drugs and individual characteristics. Another limitation of the study is that a healthy control group was not included as a third group. The presence of a healthy control group would have allowed us to evaluate the results from a broader perspective. However, the main aim of this study was to determine how disease activity in MS can be associated with markers of inflammation on the basis of NEDA. In this context, patients meeting and not meeting NEDA criteria were used as control groups against each other. 

Ratios derived from hemogram parameters such as NLR, PLR, and MLR are often used as an indicator of systemic inflammation. One of the limitations of this study is that these markers are non-specific in nature and may vary in many different pathologic conditions. However, in order to reduce this limitation while designing our study, conditions that may affect systemic inflammation parameters were determined as exclusion criteria. We would also like to point out that both groups in our study were selected homogeneously. The limitations of this study include the fact that these biomarkers were not analyzed in CSF. However, this does not invalidate the results of the study; it should only be taken into account in the interpretation of the results.

Our study is a preliminary study examining inflammatory parameters in MS based on NEDA. A key strength of this research is its focus on inflammation indexes alongside achieving 2-year NEDA-3 status. We investigated the association between the absence of disease activity and the systemic inflammatory response. Our findings suggest that systemic inflammation parameters could serve as potential biomarkers to uncover ongoing subclinical activity in the group achieving NEDA-3. This finding highlights the potential importance of considering systemic inflammation parameters. It may be useful to monitor inflammation-related biomarkers even in the absence of clear clinical signs of disease activity. 

Despite all its limitations, considering that the concept of NEDA will be questioned in depth in light of the data obtained from this study, it is thought that our study may contribute to future research in this direction. In the future, we aim to conduct a more comprehensive study on larger patient cohorts with a control group of healthy donors, addressing different clinical models of multiple sclerosis, different disease activities and other biomarkers. If these efforts lead to the discovery of inflammatory parameters with sufficiently high specificity, further studies will be needed to evaluate them not only in multiple sclerosis but also in other immune-related and neurodegenerative diseases such as Parkinson’s disease. Only then will it be possible to determine how specific and therefore useful these new markers could be in clinical practice.

## 5. Conclusions

Biomarker studies for both diagnostic and prognostic purposes in multiple sclerosis are crucial. As the medical community seeks more precise tools for diagnosis and prognosis, the value of biomarkers is becoming increasingly apparent. Recently, hemogram-derived data, which serve as a biomarker, have been increasingly utilized in various diseases. In conditions like MS, where it is challenging to assess the degree of inflammation and neurodegeneration, the significance of gleaning prognostic insights from easily accessible data in routine clinical practice is evident. This research delved into systemic inflammatory markers, such as NLR, PLR, MLR, SII, SIRI, and AISI, in MS patients. The choice of these markers stems from their potential to provide a window into the underlying inflammatory processes in MS. All of these indexes demonstrated notable correlations with physical metrics and exhibited robust interrelationships. A pivotal finding of this study is the novel revelation that SII, SIRI, and AISI can be indicators for NEDA-3 in MS. It is postulated that incorporating one or more of these non-invasive, straightforward, and practical markers with clinical and radiological criteria can enhance the monitoring of MS progression.

## Figures and Tables

**Figure 1 jpm-13-01537-f001:**
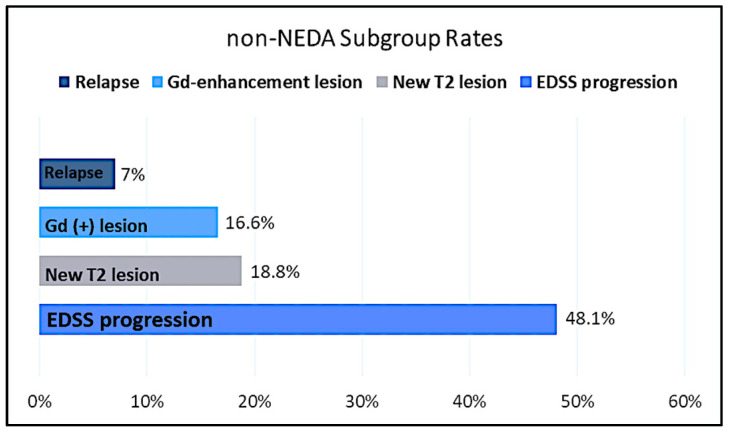
Non-NEDA subgroup rates of the study population. Gd = gadolinium; NEDA = no evidence of disease activity; EDSS = expanded disability status scale.

**Figure 2 jpm-13-01537-f002:**
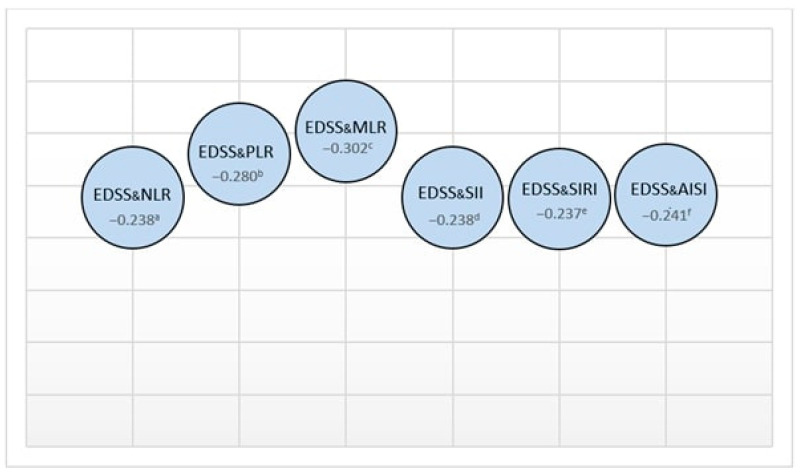
The relationship between EDSS and inflammatory indexes (rho values: a 0.014, b 0.003, c 0.002, d 0.013, e 0.014, f 0.012).

**Figure 3 jpm-13-01537-f003:**
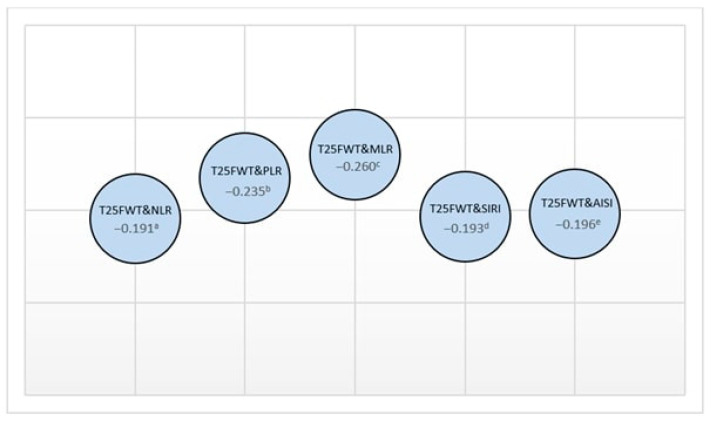
The relationship between T25FWT and inflammatory indexes (rho values: a 0.049, b 0.007, c 0.002, d 0.047, e 0.043).

**Table 1 jpm-13-01537-t001:** Formulas of inflammatory indexes.

Indexes	Formulas
NLR	Neutrophil/lymphocyte
PLR	Platelet/lymphocyte
MLR	Monocyte/lymphocyte
SII	Platelet × neutrophil/lymphocyte
SIRI	Neutrophil × monocyte/lymphocyte
AISI	Neutrophil × platelet × monocyte/lymphocyte

NLR = neutrophil-to-lymphocyte ratio; PLR = platelet-to-lymphocyte ratio; MLR = monocyte-to-lymphocyte ratio; SII = systemic immune inflammation index; SIRI = systemic immune response index; AISI = systemic total aggregation index).

**Table 2 jpm-13-01537-t002:** Indexes obtained from hemogram values and biochemical data in the study group.

	Median (Min–Max)	Mean ± SD
Monocyte (/µL)	530 (250–1340)	569.63 ± 179.94
Lymphocyte (/µL)	1108 (114–3850)	1206.06 ± 780.14
Platelet (109/L)	263 (134–469)	268.38 ± 58.65
Neutrophil (/µL)	3790 (1600–9300)	4081.11 ± 1537.32
CRP (mg/dL)	2.2 (0.4–9.9)	2.92 ± 2.28
NLR	3.33 (1.10–42.86)	6.22 ± 7.07
PLR	230.72 (82.67–2447.37)	430.12 ± 493.19
MLR	0.49 (0.13–4.56)	0.88 ± 0.94
SII	974.25 (234–11,614)	1703.98 ± 2006.43
SIRI	1.85 (0.40–24.16)	3.56 ± 4.16
AISI	506.10 (107.10–6135.90)	973.56 ± 1154.85

CRP = C-reactive protein; NLR = neutrophil-to-lymphocyte ratio; PLR = platelet-to-lymphocyte ratio; MLR = monocyte-to-lymphocyte ratio; SII = systemic immune inflammation index; SIRI = systemic immune response index; AISI = systemic total aggregation index; mean ± SD represents the average (mean) value of the dataset along with its standard deviation (SD).

**Table 3 jpm-13-01537-t003:** Relationship between the presence of NEDA and inflammatory indexes.

	NEDA-3 (+)	NEDA-3 (−)	*p*
NLR	8.99 ± 8.77	3.44 ± 2.57	<0.001
PLR	618.54 ± 610.46	241.69 ± 214.59	<0.001
MLR	1.26 ± 1.14	0.49 ± 0.42	<0.001
SII	2463.98 ± 2507.99	943.97 ± 812.67	<0.001
SIRI	5.22 ± 5.19	1.90 ± 1.51	<0.001
AISI	1426 ± 33	520.77 ± 459.56	<0.001

NLR = neutrophil-to-lymphocyte ratio; PLR = platelet-to-lymphocyte ratio; MLR = monocyte-to-lymphocyte ratio; SII = systemic immune inflammation index; SIRI = systemic immune response index; AISI = systemic total aggregation index.

**Table 4 jpm-13-01537-t004:** Correlation analysis between inflammation indexes evaluated in this study.

	rho	*p*
NLR and PLR	0.876	<0.001
NLR and MLR	0.834	<0.001
NLR and SII	0.962	<0.001
NLR and SIRI	0.908	<0.001
NLR and AISI	0.887	<0.001
SII and SIRI	0.888	<0.001
SII and AISI	0.929	<0.001
AISI and SIRI	0.969	<0.001

NLR = neutrophil-to-lymphocyte ratio; PLR = platelet-to-lymphocyte ratio; MLR = monocyte-to-lymphocyte ratio; SII = systemic immune inflammation index; SIRI = systemic immune response index; AISI = systemic total aggregation index.

**Table 5 jpm-13-01537-t005:** Results for SII, AISI, and SIRI in predicting NEDA-3.

	*p* Value	Odds Ratio (OR)	95% CI for EXP (B) (Lower–Upper)	Accuracy	Nagelkerke R Square	Cox & Snell R Square
AISI	<0.001	1.001	1.001–1.002	69.4%	0.255	0.191
SII	0.001	1.001	1.000–1.001	67.6%	0.245	0.184
SIRI	<0.001	1.511	1.216–1.877	68.5%	0.281	0.211

SII = systemic immune inflammation index; SIRI = systemic immune response index; AISI = systemic total aggregation index.

**Table 6 jpm-13-01537-t006:** Lymphocyte values according to the use of DMDs.

DMT	Lymphocyte Values (/µL)
Interferon (n = 8)	1145.50 ± 574.27
Glatiramer acetate (n = 6)	2061.67 ± 1207.36
Teriflunomide (n = 22)	1297.45 ± 859.62
Dimethylfumarate (n = 9)	1327.00 ± 748.84
Fingolimod (n = 39)	750.92 ± 418.83
Ocrelizumab (n = 5)	1491.40 ± 735.71
Cladribine (n = 6)	926.83 ± 442.33
Natalizumab (n = 3)	1700.00 ± 1131.37

DMT = disease-modifying treatment; n = number of patients.

## Data Availability

The data presented in this study are available on request from the corresponding author. The data is not available to the public due to the personal data protection law in accordance with the rules in our country.

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
