# Peer review of "A Preliminary Study on the Meaning of Inflammatory Indexes in MS: A Neda-Based Approach"

_jpm, 2023, doi:10.3390/jpm13111537_

Round 1

Reviewer 1 Report

Comments and Suggestions for Authors

This is a very important study for NEDA and the biomarkers which could be used for its prediction. Authors should include the raw values or data obtained from patient samples from CBC count and important parameter used for scoring the brain MRIs. The manuscript is not very appealing to read without any data points or relevant graphs/plots. I did not find any other concerns with the manuscript.

Author Response

For research article

Response to Reviewer 1 Comments

1. Summary

Thank you very much for your thoughtful review and valuable feedback on our manuscript. We sincerely appreciate the time and effort you dedicated to providing insights that have greatly enhanced the quality of our work.

We have meticulously addressed each of the comments and suggestions you provided. For your convenience, every single comment has been responded to line by line in the sections below. Furthermore, all the revisions and corrections made to the manuscript based on your feedback have been highlighted or tracked in the re-submitted files. This will allow for an easier identification of the changes we've made in response to your recommendations.

We believe that these revisions have strengthened our manuscript, and we hope that the changes meet your approval. Once again, thank you for your invaluable contribution to improving our work.

Thank you very much for taking the time to review this manuscript. Please find the detailed responses below and the corresponding corrections in track changes in the re-submitted files.

2. Questions for General Evaluation

Reviewer’s Evaluation

Response and Revisions

Does the introduction provide sufficient background and include all relevant references?

Yes

Are all the cited references relevant to the research?

Yes

Is the research design appropriate?

Can be improved

Thank you for pointing this out. The design of our study is centered on examining the relationship between NEDA-3, which represents the absence of disease activity in MS, and inflammation markers that are easily and practically applicable. In light of your suggestions, we have revisited the study design and have now included the DMTs that the patients are currently using. When examining the indices and lymphocyte values based on the DMTs used by the patients, we found no significant relationship between the indices and the lymphocyte count. This observation might challenge the notion that drugs causing lymphopenia can influence the inflammation index. We have highlighted this in our manuscript. We believe that with these modifications, our study design is now more robust. The relevant changes are highlighted on lines 89-91, page 2 and lines 208-216, page 7.

Are the methods adequately described?

Can be improved

In line with the valuable feedback from the reviewer, the following sentence has been added to the methods section: “NEDA's sub-parameters were defined as the presence of relapse, MRI activity (with the presence of a new T2 lesion and the presence of a gadolinium-enhancing lesion being evaluated separately), and EDSS progression.” Additionally, it has been added to the methods section that patients who have been on their current DMTs for at least one year were included in the study. The relevant changes are highlighted on lines 89-91, page 2 and lines 98-100, page 3.

Are the results clearly presented?

Can be improved

The results section has been restructured with subheadings for clearer presentation. The relevant changes are highlighted on lines 136, 144 and 153 on page 4, line 173 on page 5 and line 193 on page 6.

Are the conclusions supported by the results?

Can be improved

Thank you for pointing this out. The conclusion section has been revised and rewritten. The relevant changes are highlighted on lines 308-318, page 9.

3. Point-by-point response to Comments and Suggestions for Authors

Comments 1:

This is a very important study for NEDA and the biomarkers which could be used for its prediction. Authors should include the raw values or data obtained from patient samples from CBC count and important parameter used for scoring the brain MRIs.

Response 1: Thank you for taking the time to review our draft and for your constructive feedback. We agree with this comment.Necessary changes were made for each item in line with your suggestions. We have shared the place and explanation of these changes in the text below.

We understand the significance of including raw values and data from the CBC count, as well as the parameters used for MRI scoring.

-We have incorporated the MedianSD values for each parameter into Table 1, which pertains to the data obtained from the CBC count. Should it be necessary, we are willing to share the complete dataset with both editors and reviewers. The relevant changes are highlighted on lines 147-151, page 4.

-In response to your insightful feedback, we have detailed the subparameters of NEDA in the methods section. Specifically, we have defined and elaborated on the MRI parameters that we assessed, including the emergence of new T2 lesions and gadolinium-enhancing lesions, among others. This provides a comprehensive understanding of the MRI features considered in our study. The relevant changes are highlighted on lines 158-162, page 4.

A new sentence have been added in discussion section regarding these datas. The relevant changes are highlighted on lines 288-290, page 9.

Comments 2: The manuscript is not very appealing to read without any data points or relevant graphs/plots. I did not find any other concerns with the manuscript.

Response 2:

Thank you for your feedback regarding the inclusion of data points and visual aids in the manuscript. In response to your suggestion, we have added a figure (Figure 1) that presents the non-NEDA subgroup rates of the study population. We believe this addition will enhance the manuscript's readability and provide a clearer visual representation of the data. We appreciate your constructive comment and hope that this addition addresses your concern. The relevant changes are highlighted on lines 163-167, page 5. After adding this figure, the numbers of other figures have been revised. The relevant changes are highlighted on lines 179-181, pages 5-6.

4. Response to Comments on the Quality of English Language

Point 1: No suggestions

Response 1: none

5. Additional clarifications

Dear Reviewer,

Thank you for your involvement in the review process of our manuscript. Your feedback has played a critical role in enhancing the quality of our work. We have carefully considered each of your suggestions and have diligently made all necessary revisions. Throughout this revision process, we have strived to integrate the changes you recommended while preserving the integrity and scientific contribution of our study. We hope that the modifications and additions we've made contribute to a more comprehensive and informative manuscript.

Reviewer 2 Report

Comments and Suggestions for Authors

In this paper, Bunul and colleagues prospectively studied a cohort of 108 patients diagnosed with MS. Inflammatory markers such as NLR, PLR , MLR, SII, SIRI, and AISI were examined to determine their correlation with MS disease activity and disability. They found that SII, AISI, and SIRI were significant predictors of NEDA, and all inflammatory indexes showed a strong intercorrelation. 

The most interesting aspect of this work is certainly that the authors thought they could gain prognostic cues from data easily obtainable in normal clinical practice.

However, I believe there is a big problem. The vast majority of the DMTs taken by the patients can significantly influence all these indexes and markers and therefore this must be considered in the statistical analysis. Moreover, previous treatments may also be important. They also must be reported in a table.

Comments on the Quality of English Language

Minor issues

Author Response

For research article

Response to Reviewer 2 Comments

1. Summary

Thank you very much for your thoughtful review and valuable feedback on our manuscript. We sincerely appreciate the time and effort you dedicated to providing insights that have greatly enhanced the quality of our work.

We have meticulously addressed each of the comments and suggestions you provided. For your convenience, every single comment has been responded to line by line in the sections below. Furthermore, all the revisions and corrections made to the manuscript based on your feedback have been highlighted or tracked in the re-submitted files. This will allow for an easier identification of the changes we've made in response to your recommendations.

We believe that these revisions have strengthened our manuscript, and we hope that the changes meet your approval. Once again, thank you for your invaluable contribution to improving our work.

2. Questions for General Evaluation

Reviewer’s Evaluation

Response and Revisions

Does the introduction provide sufficient background and include all relevant references?

Must be improved

In response to your valuable suggestion, we have added two additional sentences to strengthen the introduction further. We concur with your recommendation regarding the need for a more comprehensive background and have made the necessary enhancements. We sincerely appreciate your feedback and thank you for pointing out this area of improvement. The relevant changes are highlighted on lines 63-66, page 2.

Are all the cited references relevant to the research?

Can be improved

Each of the cited references has been individually reviewed and assessed to be directly or indirectly related to the research.

Is the research design appropriate?

Must be improved

Thank you for pointing this out. The design of our study is centered on examining the relationship between NEDA-3, which represents the absence of disease activity in MS, and inflammation markers that are easily and practically applicable. In light of your suggestions, we have revisited the study design and have now included the DMTs that the patients are currently using. When examining the indices and lymphocyte values based on the DMTs used by the patients, we found no significant relationship between the indices and the lymphocyte count. This observation might challenge the notion that drugs causing lymphopenia can influence the inflammation index. We have highlighted this in our manuscript. We believe that with these modifications, our study design is now more robust The relevant changes are highlighted on lines 89-91, page 2; on lines 208-216, page 7; on lines 293-296, page 9.

Are the methods adequately described?

Must be improved

In line with the valuable feedback from the reviewer, the following sentence has been added to the methods section: “NEDA's sub-parameters were defined as the presence of relapse, MRI activity (with the presence of a new T2 lesion and the presence of a gadolinium-enhancing lesion being evaluated separately), and EDSS progression.” Additionally, it has been added to the methods section that patients who have been on their current DMTs for at least one year were included in the study. The relevant changes are highlighted on lines 89-91, page 2 and lines 98-100, page 3.

Are the results clearly presented?

Must be improved

The results section has been restructured with subheadings for clearer presentation. We believe that this format makes the content more comprehensible. The relevant changes are highlighted on lines 136, 144 and 153 on page 4, line 173 on page 5 and line 193 on page 6.

Are the conclusions supported by the results?

Must be improved

Thank you for pointing this out. The conclusion section has been revised and rewritten The relevant changes are highlighted on lines 308-318, page 9.

3. Point-by-point response to Comments and Suggestions for Authors

Comments 1: In this paper, Bunul and colleagues prospectively studied a cohort of 108 patients diagnosed with MS. Inflammatory markers such as NLR, PLR , MLR, SII, SIRI, and AISI were examined to determine their correlation with MS disease activity and disability. They found that SII, AISI, and SIRI were significant predictors of NEDA, and all inflammatory indexes showed a strong intercorrelation. 

The most interesting aspect of this work is certainly that the authors thought they could gain prognostic cues from data easily obtainable in normal clinical practice.

However, I believe there is a big problem. The vast majority of the DMTs taken by the patients can significantly influence all these indexes and markers and therefore this must be considered in the statistical analysis. Moreover, previous treatments may also be important. They also must be reported in a table.

Response 1: Firstly, we would like to express our agreement with the concerns you raised regarding the potential influence of DMTs on the inflammatory indexes and markers. We recognize the validity of your point. However, our detailed explanation on this matter is provided below.

In patients being monitored with a diagnosis of multiple sclerosis, those not on medication are typically in the early stages of the disease. For NEDA-based studies, there is a specific follow-up period required to assess whether the NEDA status is achieved, and at this stage, the proportion of untreated patients is quite low. For this reason, our study included patients who are currently on DMTs.

To address concerns, we have stratified the patients based on the DMTs they were taking. Given that some DMTs, especially Fingolimod, have a known potential to cause lymphopenia, we have added a table (Table 6) that presents lymphocyte values for each DMT group. All DMT subgroups were evaluated using the Kruskal-Wallis test, and no significant differences were observed among the inflammatory markers across drug subgroups, including those that can cause lymphopenia The relevant changes are highlighted on lines 208-216 , page 7 and on lines 293 – 296, page 9. This suggests that the DMTs taken by the patients may not significantly skew our findings related to the inflammatory markers.

Regarding your valuable suggestion on previous treatments, we have focused on patients who have been on their current medication for at least one year. Patients whose medications were changed in the last 1 year were not included in the study. The relevant changes are highlighted on lines 89-91, page 2.  

4. Response to Comments on the Quality of English Language

Point 1: Minor editing of English language required

Response 1: Our manuscript has been revised in line with your suggestions, and we have engaged professional services for English editing. The certificate of this service has also been attached.

5. Additional clarifications

Dear Reviewer,

Thank you for your involvement in the review process of our manuscript. Your feedback has played a critical role in enhancing the quality of our work. We have carefully considered each of your suggestions and have diligently made all necessary revisions. Throughout this revision process, we have strived to integrate the changes you recommended while preserving the integrity and scientific contribution of our study. We hope that the modifications and additions we've made contribute to a more comprehensive and informative manuscript.

Reviewer 3 Report

Comments and Suggestions for Authors

The manuscript ‘’The Meaning of Inflammatory Indexes in MS: 2 Neda-Based Approach’’ reflects the relationship between inflammatory indexes and MS disease activity and progression. The authors have revised various inflammation indexes derived from hemogram parameters, such as NLR, PLR, and MLR. The subject and results of the article is under significant scientific importance, however the part of Results description might be restructured. We advise to divide the description of the results into separate parts, for a better understanding, for instance:

3.1 Hemogram values and biochemical data obtained from study group

3.2 …

 Grammar and text layout also should be revised carefully, there are numerous errors, for instance:

Line 78: Participitians:,  - Participants

Line 167, 168, 169 and 170 are empty.

Conclusion:

This version of the article is suitable for scientific publication, after revision of description of results and grammar.

Comments on the Quality of English Language

This version of the article is suitable for scientific publication, after revision of description of results and grammar.

Author Response

For research article

Response to Reviewer 3 Comments

1. Summary

Thank you very much for your thoughtful review and valuable feedback on our manuscript. We sincerely appreciate the time and effort you dedicated to providing insights that have greatly enhanced the quality of our work.

We have meticulously addressed each of the comments and suggestions you provided. For your convenience, every single comment has been responded to line by line in the sections below. Furthermore, all the revisions and corrections made to the manuscript based on your feedback have been highlighted or tracked in the re-submitted files. This will allow for an easier identification of the changes we've made in response to your recommendations.

We believe that these revisions have strengthened our manuscript, and we hope that the changes meet your approval. Once again, thank you for your invaluable contribution to improving our work.

Thank you very much for taking the time to review this manuscript. Please find the detailed responses below and the corresponding corrections in track changes in the re-submitted files.

2. Questions for General Evaluation

Reviewer’s Evaluation

Response and Revisions

Does the introduction provide sufficient background and include all relevant references?

Yes

Are all the cited references relevant to the research?

Yes

Is the research design appropriate?

Yes

Are the methods adequately described?

Are the results clearly presented?

Can be improved

The results section has been restructured with subheadings for clearer presentation. The relevant changes are highlighted on lines 136, 144 and 153 on page 4, line 173 on page 5 and line 193 on page 6.

Are the conclusions supported by the results?

Yes

3. Point-by-point response to Comments and Suggestions for Authors

Comments 1: The manuscript ‘’The Meaning of Inflammatory Indexes in MS: 2 Neda-Based Approach’’ reflects the relationship between inflammatory indexes and MS disease activity and progression. The authors have revised various inflammation indexes derived from hemogram parameters, such as NLR, PLR, and MLR. The subject and results of the article is under significant scientific importance, however the part of Results description might be restructured. We advise to divide the description of the results into separate parts, for a better understanding, for instance:

3.1 Hemogram values and biochemical data obtained from study group

3.2 …

Response 1: Thank you for your valuable feedback on our manuscript. In line with your suggestions, we have restructured the Results section with subheadings for clearer presentation. We believe that this format enhances the comprehensibility of the article. The results section has been restructured with subheadings for clearer presentation. The relevant changes are highlighted on lines 136, 144 and 153 on page 4, line 173 on page 5 and line 193 on page 6.

Comments 2: Grammar and text layout also should be revised carefully, there are numerous errors, for instance:

Line 78: Participitians:,  - Participants

Line 167, 168, 169 and 170 are empty.

Response 2:  Thank you for pointing out the errors in our manuscript. We have obtained professional editing services for grammar, and the certificate of this service has been attached. The mentioned lines have been corrected, and we have thoroughly reviewed the entire document to ensure no empty lines remain. We appreciate your attention to detail and your constructive feedback.

4. Response to Comments on the Quality of English Language

Point 1: This version of the article is suitable for scientific publication, after revision of description of results and grammar.

Response 1: Our manuscript has been revised in line with your suggestions, and we have engaged professional services for English editing. The certificate of this service has also been attached.

5. Additional clarifications

Dear Reviewer,

Thank you for your involvement in the review process of our manuscript. Your feedback has played a critical role in enhancing the quality of our work. We have carefully considered each of your suggestions and have diligently made all necessary revisions. Throughout this revision process, we have strived to integrate the changes you recommended while preserving the integrity and scientific contribution of our study. We hope that the modifications and additions we've made contribute to a more comprehensive and informative manuscript.

Round 2

Reviewer 2 Report

Comments and Suggestions for Authors

no further comments